# Machine Learning Approaches to Identify Discriminative Signatures of Volatile Organic Compounds (VOCs) from Bacteria and Fungi Using SPME-DART-MS

**DOI:** 10.3390/metabo12030232

**Published:** 2022-03-08

**Authors:** Mehak Arora, Stephen C. Zambrzycki, Joshua M. Levy, Annette Esper, Jennifer K. Frediani, Cassandra L. Quave, Facundo M. Fernández, Rishikesan Kamaleswaran

**Affiliations:** 1School of Electrical and Computer Engineering, Georgia Institute of Technology, Atlanta, GA 30332, USA; 2Department of Biomedical Informatics, Emory University School of Medicine, Atlanta, GA 30332, USA; rkamaleswaran@emory.edu; 3School of Chemistry and Biochemistry, Georgia Institute of Technology, Atlanta, GA 30332, USA; zambrzyck@musc.edu (S.C.Z.); facundo.fernandez@chemistry.gatech.edu (F.M.F.); 4Department of Otolaryngology—Head and Neck Surgery, Emory University School of Medicine, Atlanta, GA 30332, USA; joshua.levy2@emory.edu; 5Division of Pulmonary, Allergy, Critical Care, and Sleep Medicine, Emory University School of Medicine, Atlanta, GA 30332, USA; aesper@emory.edu; 6Emory Critical Care Center, Emory University School of Medicine, Atlanta, GA 30332, USA; 7Nell Hodgson Woodruff School of Nursing, Emory University, Atlanta, GA 30332, USA; jfredia@emory.edu; 8Department of Dermatology, Emory University School of Medicine, Atlanta, GA 30332, USA; cassandra.leah.quave@emory.edu; 9Center for the Study of Human Health, Emory College of Arts and Sciences, Atlanta, GA 30332, USA; 10Department of Emergency Medicine, Emory University School of Medicine, Atlanta, GA 30332, USA

**Keywords:** DART-MS, ambient plasma ionization, solid phase micro-extraction, machine learning classification algorithms, K-means clustering, imbalanced learning, VOC, point-of-care devices, pathogen identification

## Abstract

Point-of-care screening tools are essential to expedite patient care and decrease reliance on slow diagnostic tools (e.g., microbial cultures) to identify pathogens and their associated antibiotic resistance. Analysis of volatile organic compounds (VOC) emitted from biological media has seen increased attention in recent years as a potential non-invasive diagnostic procedure. This work explores the use of solid phase micro-extraction (SPME) and ambient plasma ionization mass spectrometry (MS) to rapidly acquire VOC signatures of bacteria and fungi. The MS spectrum of each pathogen goes through a preprocessing and feature extraction pipeline. Various supervised and unsupervised machine learning (ML) classification algorithms are trained and evaluated on the extracted feature set. These are able to classify the type of pathogen as bacteria or fungi with high accuracy, while marked progress is also made in identifying specific strains of bacteria. This study presents a new approach for the identification of pathogens from VOC signatures collected using SPME and ambient ionization MS by training classifiers on just a few samples of data. This ambient plasma ionization and ML approach is robust, rapid, precise, and can potentially be used as a non-invasive clinical diagnostic tool for point-of-care applications.

## 1. Introduction

Timely and accurate detection of pathogens in the human body can play a critical role in the early detection of infection, greatly improving prognosis and recovery [1]. Conventional methods for the identification of bacteria and other pathogens are either very slow, such as microbial culturing, or require highly specialized equipment and expert knowledge, such as molecular identification [2,3]. At present, there is a need for rapid, sensitive, low-cost, and non-invasive screening tools that can be deployed as point-of-care (POC) devices [4]. These POC tools employed in the proximity of patient care are essential to expedite patient care and decrease reliance on slower diagnostic techniques to identify pathogens and their associated antibiotic resistance [4]. Many efforts are underway to develop new point-of-care tools such as Raman spectroscopy and gas chromatography [5,6,7].

Volatile organic compounds are a diverse family of organics that microbial pathogens produce as a result of their metabolic activity [8]. Studies have shown that the VOC profiles emitted by microbes vary with species, environmental conditions, and ambient factors [9]. Identification of pathogens through their VOC profiles by mass spectrometry (MS) has seen extensive research [1,2,10,11]. These studies have analyzed VOCs emissions in exhaled breath, blood, and urine, and have been successful in demonstrating the diagnostic potential of bacterial VOC profiling [12,13,14]. Exhaled breath analysis (EBA) has been extensively studied for disease monitoring [15,16]. Skin VOC sampling and analysis have also picked up pace, and various studies have shown its effectiveness in identifying biomarkers for physiological processes [17,18,19]. The work proposed in this study builds on this premise and presents a method for identifying a pathogen based on its VOC profile.

Human skin hosts a unique microbiome. The perfusion of gases from the skin through surface capillaries makes it an important source of emission of VOCs from the human body [20]. Recently, there have been studies that analyze VOCs collected from the skin to determine surface acidity [17], identify biomarkers of physiological reactions [18], and detect the presence of certain diseases [19]. The field of VOC sensing to detect and monitor the diversity of human disease has thus far been limited by an absence of POC technologies leveraging machine learning (ML) based diagnostic VOC signatures. Fitzgerald et al. [1] have shown that it is possible to discriminate between bacterial strains from VOC profiles obtained through headspace SPME gas chromatography MS; however, this technique relies on lengthy separations that make it practical.

The most common method for VOC profiling is via multivariate analysis, such as principal component analysis [21,22]. In recent years, applications of machine learning (ML) and artificial intelligence have shown great promise in advancing the field of healthcare and critical care [23]. ML models are able to identify physiomarkers that help in early detection of sepsis [24] and predict life-threatening conditions such as acute respiratory distress syndrome (ARDS) using ICU data [25] and gene expression signatures [26]. A major area of research currently is early and accurate detection of infections from microbial VOCs and several statistical and machine learning methods have been successfully developed to this end [4,27]. There are works that demonstrate the use of ML for VOC screening via gas identification using ”e-nose” metal-oxide sensors [28,29]. Palma et. al. show that supervised learning can be successfully used to classify microbial strains using meta-information about their VOC profiles [30]. Other works add to this by showing that SPME and gas chromatography mass spectrometry can be used to sample VOC signatures, which can be used to create VOC profiles for classification [31,32,33]. In this work, we use signal processing and ML techniques to develop a rapid, robust and end-to-end Python pipeline for classifying a pathogen as bacteria or fungi, using the raw MS spectrum.

In this work, solid phase micro-extraction (SPME) and direct analysis in real time (DART) ambient plasma ionization MS were used to collect VOC sample data from microbial cultures followed by automatic feature extraction and learning techniques for VOC profiling. VOC fingerprints have been shown for fungi with DART alone [34] and SPME-DART has been previously reported for the analysis of VOC in samples such as grapes [35] and museum pieces [36]. This study involves using SPME to collect VOCs accumulated in the gas space above the agar slant in a sealed tube, known as the headspace. The SPME blades were then transported to be rapidly processed by DART and high-resolution MS. The VOC samples were profiled as bacteria or fungi and further analyzed to identify individual strains of bacteria. This method has the potential to improve patient care by rapidly identifying human pathogens at the POC by future applications in skin VOC sampling. High-resolution MS was used in this initial discovery phase and could potentially be replaced by portable MS instruments coupled to DART in POC settings to reduce cost.

## 2. Results and Discussions

### 2.1. Classification of Pathogens as Bacteria or Fungi Based on VOC Signatures

The MS samples were sent through a blank correction, smoothing, and peak detection pipeline to obtain a binary feature matrix with 1’s indicating peak locations. Principal Component Anaysis (PCA) was performed on this binary feature matrix, before the supervised learning stage. Unsupervised K-means clustering on PCA-transformed data (Figure 1) showed that all fungi were correctly clustered together, but some bacterial samples were assigned to the fungi cluster. The fact that an unsupervised clustering algorithm could identify underlying structure and pattern in the data that can be leveraged to broadly identify the two pathogen types laid a promising premise for further experiments with supervised learning algorithms. The support vector machine binary classifier, K nearest neighbor classifier, and a logistic regression classifier could differentiate between bacterial and fungal pathogens with high accuracy. Decision tree classifiers provided interpretability in terms of identifying salient features that can help distinguish between bacteria and fungi with high accuracy (Figure 2c). Results for the performance of these supervised learning algorithms on the original binary feature matrix as well as the PCA-transformed dataset are listed in Table 1.

Training the classifiers directly on the binary feature matrix obtained after peak detection helped determine exactly which m/z values the classifier algorithms were used to discriminate between bacteria and fungi. This pipeline involved less feature engineering and led to more interpretable results. Random forest classifiers performed significantly better when trained on the original binary feature matrix. It was also easier to visualize the salient peak locations that had maximum discriminatory importance. Each ML model was compared to each other using receiver operating characteristic curves (ROC) and precision–recall (PRC) curves. ROC curves display the sensitivity (true positive rate) and specificity (1-false positive rate) tradeoff of a classifier. PRC plots the precision (positive prediction power) vs. recall (sensitivity). ROC and PRC results for supervised learning algorithms are shown in Figure 3. The area under the curve (AUC) can be treated as an estimate of how well a model performs. All the ML approaches were verified by 3-fold cross-validation. It can be seen that random forest classifiers trained on the binary feature matrix had the best AUC, while SVM performed well on the PCA feature matrix. Logistic regression performed consistently well on both datasets, as can also be verified from Table 1.

### 2.2. Classifying Individual Bacterial Strains

The dataset consisted of 10 strains of bacteria, with three samples per strain. The training set consisted of two samples per strain, while the third sample was placed in the test set. Even with this limited data set, the decision tree classifiers were able to classify 4 out of the 10 strains in the test set accurately, while the SVM classifier was able to classify 5 out of the 10 strains correctly. Two samples in the training set for each class limited variability introduced by the up-sampling with SMOTE to train the learning models and avoid bias. Further work with larger sample sets needs to be conducted to expand the dataset and validate the proposed method. The observed results show the potential of this data processing workflow in pathogen identification. Certain strains of bacteria were consistently classified correctly with a high degree of confidence by all the trained classifiers. Interpreting the features of discriminatory importance from the decision tree classifiers trained on the normalized and blank corrected MS data could help determine the peaks used for discriminating *Proteus mirabilis* from the other strains of bacteria highlighted in Figure 4.

### 2.3. Discussions

While many reports have used data analytics, dimensionality reduction, and clustering techniques to study mass spectra and identify useful markers, very few have focused on the use of supervised ML algorithms for VOC profiling. One reason for this lack of success is the lack of sufficient data to effectively train these supervised learning algorithms. In this study, we used the synthetic minority oversampling technique (SMOTE) [37] to artificially up-sample data by populating each class in the training set with convex combinations of existing data points, ensuring that the new data points retain similar characteristics. Our artificial upsampling and learning pipeline that makes it possible to train machine learning algorithms on a few samples of SPME-DART-MS data, with high precision. Most previous work [30,31,32,33] rely on an intermediate step of peak detection and identification of VOCs using the NIST05 mass spectral library. In our work, an end-to-end pipeline is developed using python, which takes the raw MS data as input and automatically performs background subtraction, peak detection, feature identification, and classification.

Further analyzing the salient discriminatory features that were used for classification shows that the ML algorithms are able to automatically identify peak locations in the MS data that have discriminatory importance. Corresponding VOCs can be identified using this information. This works marks progress in the field of data analytics and processing for metabolite detection and identification, as it not only describes a fully automated method to discriminate between pathogens using VOC signatures but also provides insight into the salient discriminatory features of the VOC signatures so that the results can be interpreted in a biological and chemical context.

To realize the goal of effective point-of-care diagnostics, it is necessary to develop rapid, robust, and precise methods for the detection and diagnosis of these infectious diseases [4]. This study describes a proof-of-principle method for pathogen identification via VOC profiling using DART-SPME samples as a quick and accurate tool for detection of infection. The novelty of this work lies in using SPME and DART ambient plasma ionization MS to rapidly acquire VOC signatures of bacteria and fungi from human skin, designing a data preprocessing pipeline for the MS data using SMOTE to tackle class imbalance, and training ML algorithms successfully in a low-data regime to achieve a precision of 0.92 for identifying bacteria and a precision of 0.72 for identifying fungi.

The results show that supervised classifiers, trained on as few as 40 samples of data, are able to differentiate between the two classes of pathogens with an accuracy close to 90%. With access to more data, it will likely be possible to extend the capabilities of this model such that it can identify the type of pathogen and the specific strains of bacteria and fungi. For this proof-of-principle study, we have limited our analysis to PCA followed by supervised classification. The advantage of PCA is that it is quick, stable, and able to retain interpretability in terms of identifying the peak locations in the MS data that have discriminatory importance. A possible direction for future work would be to explore other classes of multivariate analysis, in particular, Bray–Curtis dissimilarity with NMDS (non-metric multidimensional scaling), which performs well on sparse data. It would be interesting to see whether it can improve the identification of specific bacteria and fungi strains.

Another limitation of this study was the lack of access to an external VOC database to compare the accurate masses to. This could be used to perform structural elucidation on discriminatory features. For the purpose of this study, the pathogen VOC signatures were collected in a controlled laboratory environment. Future work would entail shifting to real-world settings and sampling pathogen VOCs directly from the skin surface of patients.

## 3. Materials and Methods

### 3.1. Sample Preparation

Ten species of human pathogenic bacteria and three fungi were incubated at 37 ${}^{\circ }$C overnight on tryptic soy agar slants and sabouraud agar slants, respectively. Visible streaks of microbial growth were observed on the agar surface. A tryptic soy agar and a sabouraud agar slant with no microbes were used as a blank reference headspace. Each species of bacteria and fungus used in this study is listed in Table 2. SPME of the slant headspace was conducted using coated blade spray (CBS) blades with a hydrophilic-lipophilic coating (Restek; Bellefont, PA, USA). Prior to headspace collection, blades were washed with high purity liquid chromatography MS-grade methanol from Fisher Scientific (Hampton, NH, USA) and air dried. Three SPME blades were inserted and sealed into each of the 15 slants in this study. Each set of blades in the sealed slant was incubated at room temperature for 5 min. Blades were then removed after incubation and immediately stored in individual microcentrifuge tubes until processing.

### 3.2. Ambient Plasma Ionization Mass Spectrometry

VOCs collected on SPME blades were desorbed and ionized by a DART simplified voltage and pressure (SVP) ambient plasma ionization source from Ionsense (Saugus, MA, USA). DART operates through a combined thermal desorption and metastable-induced chemical ionization mechanism. A full description of the DART operation can be found elsewhere [15]. Ultrahigh-purity helium from Airgas (Atlanta, GA, USA) was used to sustain the DART plasma which was heated to 200 ${}^{\circ }$C by the DART’s resistive element. The DART source was coupled to a Waters (Milford, MA, USA) Synapt G2-S via Ionsense’s VAPUR interface for discovery and method optimization experiments. DART-MS experiments were conducted with a 1 s scan time in time-of-flight-only sensitivity mode. The acquisition range was 50–700 m/z in positive ion mode. The Synapt’s ion source inlet temperature was set to 100 ${}^{\circ }$C. The sample cone and source offset were set to 30 V and 50 V, respectively. The Restek CBS blades were placed between the DART plasma stream and the mass spectrometer inlet as shown in Figure 2a. Briefly, a coated blade was removed from the microcentrifuge tube, mounted to a linear rail attached to the VAPUR interface, and then slid into position between the DART plasma exit and the mass spectrometer atmospheric pressure inlet. The flat nature of the CBS substrate ensured that the fluid dynamics were favorable for effective ion injection [38]. After desorption of VOCs from the blade by DART for 60 s, the blade was retracted and a new blade was mounted on the rail for conducting the next analysis.

### 3.3. Data Preprocessing

The data processing pipeline is illustrated in Figure 2b. First, mass spectra were normalized by subtracting the mean and dividing by the absolute maximum. Next, linear interpolation was applied to the pathogen data to align the m/z axis with the blank spectra for computation. The blank was then subtracted from the pathogen data. The blank-subtracted pathogen spectra were then smoothed by a mean filter. Adaptive thresholding was applied to windows of 50 m/z to gather peak locations. These peak locations were encoded into a binary feature matrix that indicated whether a peak was present in a given m/z interval. The data processing and subsequent classification were conducted using Python in the Jupyter Notebook environment. All code is shared in a publicly available GitHub Repository.

Since the dataset was limited to 39 samples and heavily imbalanced towards the bacterial class, an artificial up-sampling technique called synthetic minority oversampling technique (SMOTE) [37] was utilized to generate a new up-sampled training set for the class with the lowest number of objects. This ensures ML models are more robust and have better predictive abilities. The new data points were created by SMOTE through random convex combinations of existing data points in feature space. This approach is useful while training ML algorithms in low-data or imbalanced class scenarios. The open-source imbalanced-learn library’s implementation of SMOTE [39] was used to up-sample this dataset. Figure 5 shows Principal Component Analysis (PCA) on (a) original VOC data, and (b) the dataset after SMOTE up-sampling.

### 3.4. Machine Learning Classification Algorithms

The primary objective of this work was to automatically classify a pathogen as bacteria or fungi using its VOC mass spectrum. Automatic identification of specific bacterial strains was explored as a secondary objective. The preliminary dataset consisted of 39 data points. Each data point corresponded to the binary feature vector obtained after peak detection. A train:test ratio of 7:3 was used to split the data, and the training set was populated using SMOTE. The final unsampled dataset consisted of 84 train cases (42 bacteria and 42 fungi) and 36 test cases (18 bacteria and 18 fungi). Principal component analysis (PCA) was applied on this up-sampled dataset. The top 15 principal components were used as input into the ML models. This number was decided by analyzing the elbow plot of explained variance ratio to the number of components. Five different ML classification algorithms were trained and tested for the classification task, implemented using Python’s scikit-learn library. These five ML algorithms included:
Logistic Regression with ’L2’ regularization (Ridge Regression): This is a simple linear classification model that achieves good performance for linearly separable classes. A binary classifier was implemented with the stochastic average gradient (SAG) solver and regularized with an ‘L2’ prior.Logistic Regression with ’L1’ regularization (Lasso Regression): This is also a linear model that promotes sparsity in the learnable parameters that are can be seen as weights for each variable. The classifier was implemented with the stochastic average gradient (SAG) solver and regularized with an ‘L1’ prior.Decision Trees and Random Forests: The decision tree algorithm learns to predict the class of a given input by a series of simple decision rules that are inferred from the training data. Random forests are ensemble classifiers that train multitudes of decision trees on different subsets of features, each being trained on a bootstrapped subset of the training data. A random forest classifier was also trained on the PCA-transformed data, as well as on the binary feature matrix. A huge advantage of these methods is that they help identify subsets of input variables that may be most or least relevant to the problem. In our case, we can see the exact peak locations that were of discriminatory importance to the classifier (Figure 2b).Support Vector Machines (SVM): A support vector machine classifier works by finding a classification boundary that best separates the data points in the training set. It is not limited to finding a linear model and is able to find optimal separation in higher dimensional subspaces. An SVM classifier was trained on both the PCA-transformed data, as well as the binary feature matrix. A 5-fold cross-validation-based grid search was used to choose between linear kernels, radial basis function (rbf) kernels, and sigmoid kernels, as well as to choose the optimal hyperparameters.K-Nearest Neighbors (KNN): The K-nearest neighbors algorithm classifies a new data point by simply considering the class of a certain *k* number of data points in the training set that lie closest to it in the feature space, and then choosing the most frequently occurring class label. A KNN classifier was trained using 8 nearest neighbors, this being determined using a 5-fold cross-validation-based grid search.

The methods listed above are supervised learning methods, i.e., the algorithm uses knowledge of class labels of data points in the training set to learn parameters or rules for predicting the class of a new, unlabeled data point. We also experiment with K-means clustering, an unsupervised learning algorithm, which clusters data points that are most similar (according to a set of salient features that are learned by the algorithm). The algorithm attempts to find structure or patterns in the data without using explicit data labels (bacteria/fungi). Results of the clustering are depicted in Figure 1.

## 4. Conclusions

This study presents a new approach to identify pathogens from volatile organic compound (VOC) signatures collected from the skin, that is robust, rapid, and precise, and can potentially be used as a non-invasive clinical diagnostic tool for point-of-care applications.

## Figures and Tables

**Figure 1 metabolites-12-00232-f001:**
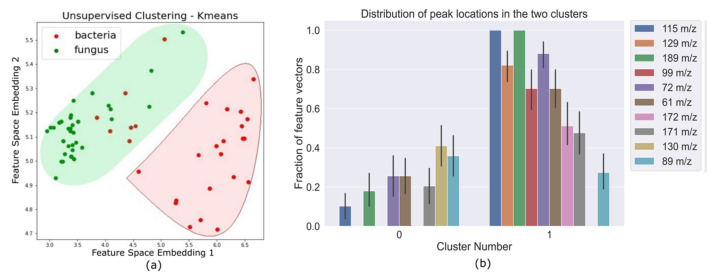
(**a**) Feature vectors corresponding to each pathogen as data points in the transformed feature space after k-means clustering for two clusters. The colored shaded regions cover all the points that were clustered together and the red and green differentiate between the two clusters. We can see that there is some inherent separability between bacteria (red data points) and fungi (green data points) in this transformed feature space. (**b**) This plot shows the variability of salient peak locations in each cluster. The clustering algorithm is able to automatically identify peak locations that are commonly seen in one pathogen type, and not seen in the other.

**Figure 2 metabolites-12-00232-f002:**
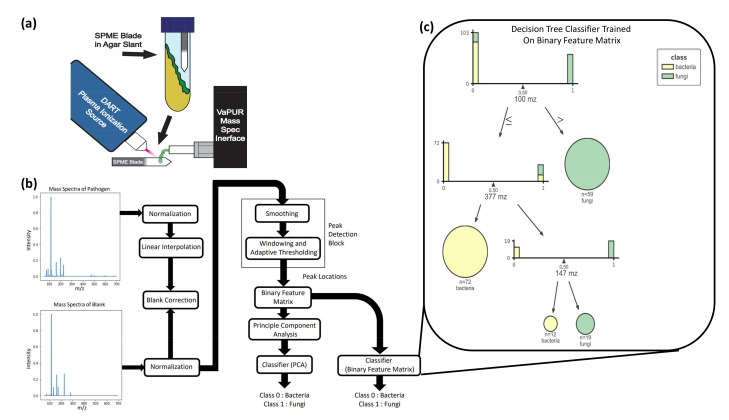
Study overview for identification of pathogens from skin VOCs. (**a**) Summary diagram of the SPME-DART-MS procedure. SPME blades are incubated in the headspace of the agar slant with the microbes. The blades are then removed and placed in between the plasma stream of the DART plasma ionization source and the ambient pressure interface of the mass spectrometer. VOC adhered to the SPME blade are desorbed and ionized by DART. Then, the ionized VOC enter the ambient pressure interface of the mass spectrometer for measurement to produce a signature. (**b**) Process flow diagram for MS data preprocessing, peak detection, and ML. The mass spectra of the pathogen and blank were first min-max normalized. Linear interpolation was applied to the pathogen data to match the m/z sampling frequency of the blank to facilitate the blank subtraction step. The pathogen mass spectra were then smoothed by a mean filter, after which adaptive thresholding was applied to windows of 50 m/z intervals to obtain peak locations. These are encoded into a binary feature matrix that indicates whether a peak is present in a unit m/z interval or not. A PCA transform on this matrix was used as the input to our ML classification algorithms. (**c**) Interpretation of the decision tree trained on the binary feature matrix in terms of feature importance. This figure shows the distribution of bacteria and fungi samples with respect to the peak locations of most discriminatory importance (100 m/z, 377 m/z, 147 m/z).

**Figure 3 metabolites-12-00232-f003:**
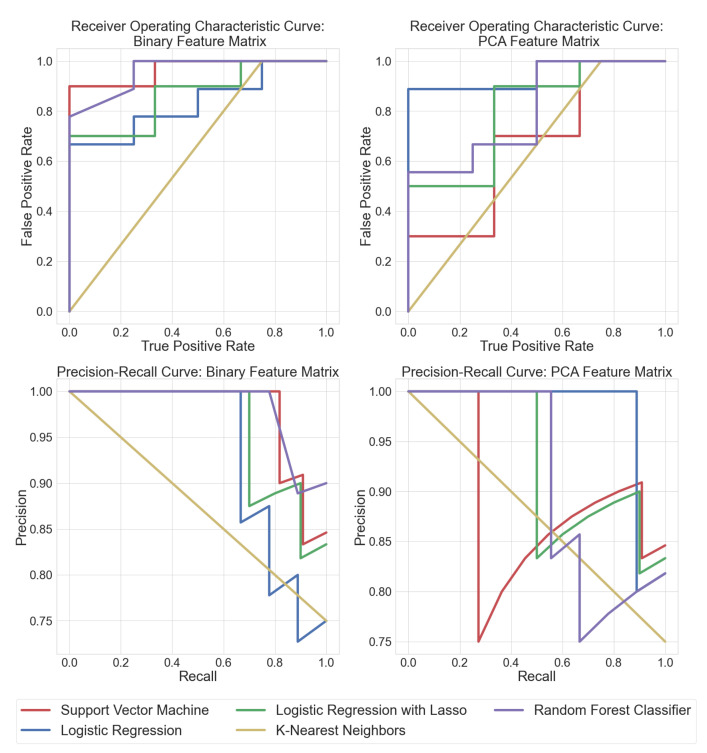
Receiver operating characteristic curves (ROC) depict the tradeoff between true positives vs. false positives for each algorithm (traces rising closer to the top left are better). The precision–recall curves (PRC) show the tradeoff between true positive vs precision in positive predictions for each algorithm (traces closer to the top right are better). This figure displays the ROC and PRC curves for classifiers trained on the binary feature matrix as well as the feature matrix after PCA, respectively.

**Figure 4 metabolites-12-00232-f004:**
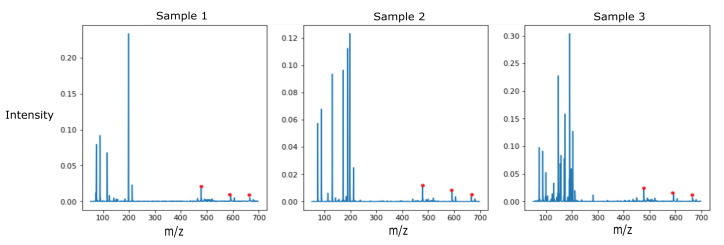
Detected VOCs at m/z 478.3883, m/z 592.2332, and m/z 666.2437 for all 3 samples of *Proteus mirabilis* (CDC-0029).

**Figure 5 metabolites-12-00232-f005:**
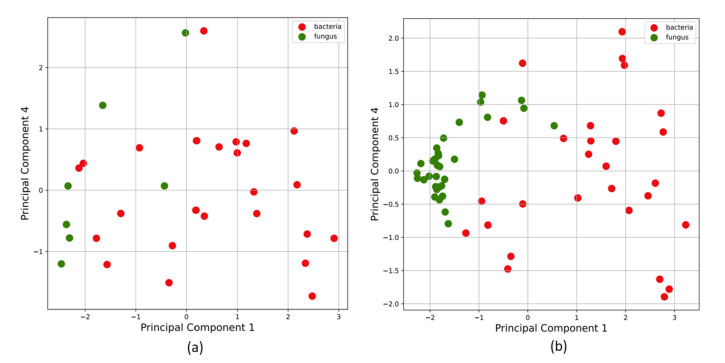
Principal component analysis on (**a**) original VOC data, and on (**b**) the dataset after SMOTE upsampling. SMOTE is a dataset upsampling technique that was used in this study to add artificial datapoints generated via random convex combinations of existing data points in the feature space. This is useful while training ML algorithms on class-imbalanced datasets and low-data regimes. In our study, the bacteria class (Class 0) had 30 samples and the fungi class (Class 1) had 9 samples. SMOTE was used to combat the limited dataset and class imbalance issues to train our ML algorithms with lesser bias.

**Table 1 metabolites-12-00232-t001:** Results of binary classification (bacteria vs. fungi) for the five supervised ML algorithms (averaged over 3-folds) on the binary feature matrix and the PCA-transformed data. The numbers in bold mark the highest values for each metric. It can be observed that decision tree classifiers trained on the binary feature matrix perform well, with highest overall accuracy, f-score, and AUC.

Classifier	Dataset	Accuracy	F-Score	Area under the ROC Curve (AUC)	Class Bacteria		Class Fungi	
	Precision	Sensitivity	Precision	Sensitivity				
Logistic Regression	Binary Features	0.846	0.748	0.865	0.903	0.899	0.639	0.667
	PCA Features	0.846	0.843	0.775	0.853	0.970	0.833	0.444
Logistic Regression with Lasso	Binary Features	0.795	0.753	0.921	**0.928**	0.870	0.633	**0.722**
	PCA Features	0.795	0.742	0.827	0.895	0.870	0.633	0.639
K-Nearest Neighbors	Binary Features	0.821	0.782	0.743	0.886	0.903	0.700	0.583
	PCA Features	0.820	0.812	0.752	0.886	0.936	0.750	0.583
Support Vector Machines	Binary Features	0.795	0.657	0.805	0.870	0.862	0.528	0.583
	PCA Features	0.821	0.670	0.734	0.842	0.899	0.556	0.444
Random Forest Classifier	Binary Features	**0.872**	**0.937**	**0.951**	0.881	**0.982**	**0.980**	0.555
	PCA Features	0.744	0.570	0.779	0.794	0.903	0.444	0.194

**Table 2 metabolites-12-00232-t002:** This table lists the specific bacteria and fungus species evaluated using SPME-DART-MS in this study.

Pathogen	Strain
	*Staphylococcus aureus* (LAC)
	*Staphylococcus aureus* (UAMS-1)
	*Staphylococcus epidermidis* (NRS-101)
	*Acinetobacter baumannii* (CDC-0033)
Bacteria	*Klebsiella pneumoniae* (CDC-0004)
	*Pseudomonas aeruginosa* (PA01)
	*Klebsiella aerogenes* (NR-48555)
	*Enterococcus faecium* ( HM-959 )
	*Escherichia coli* (CDC-0346)
	*Proteus mirabilis* (CDC-0029)
	*Candida albicans* ( NR-29340 )
Fungi	*Candida glabrata* (CDC-0314)
	*Malassezia fufur* (ATCC-12078)

## Data Availability

The data presented in this study are publicly available at https://github.com/rkamaleswaran/VOCpath (accessed on 6 March 2022).

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
