# Peer review of "Machine Learning Approaches to Identify Discriminative Signatures of Volatile Organic Compounds (VOCs) from Bacteria and Fungi Using SPME-DART-MS"

_metabolites, 2022, doi:10.3390/metabo12030232_

Round 1

Reviewer 1 Report

This manuscript summarizes the development of a statistical model using the detection of VOCs to ascertain specific communities of pathogenic bacteria and fungi using SPME-DART-HRMS. The manuscript is well written with few grammar errors and the work is original and unique. As a proof of concept method development study, this is a well-designed and presented study. I look forward to seeing the application of the method to real-world samples which will require more nuanced development strategies.

  • Have you used other coated blades for SPME procedure? If so, have you noticed any effect in the analytes collected between species?
  • Did you notice an effect on desorption time and number of unique species identified or sensitivity?
  • Why did you choose to use PCA to show the relationships between fungus and bacterial species instead of Bray-Curtis similarity using NMDS? I imagine some null values exist for some of the species identified, for which using Euclidean distance may not be suitable.
  • Did you perform any structure elucidation on the significant VOCs that distinguish species or between bacteria and fungi or did you rely on the full scan exact mass for any identifications? It would be more interesting to see some data regarding what VOCs are unique to bacteria or fungi and among different subcommunities. For instance in Figure 5 you have three unique m/z’s that you identified across three SPME replicates, did you make a molecular identification for these?
  • Can you comment on the variability observed between the three blades for each sample?
  • Line 169: I believe you mean to use the word “closest” instead of “closet”.
  • You list “Skin VOCs” as a potential keyword. However in your study you never explored the detection of VOCs present on skin or simulated skin, rather your collections were done from agar tubes. It is unknown if the VOCs (and thus the model you build) will be able to perform with communities of bacteria on other surfaces such as skin).

Reviewer 2 Report

Authors used the SPME and ambient plasma ionization mass spectrometry to rapidly acquire volatile organic compounds signatures of bacteria and fungi. According to the authors, this ambient plasma ionization and machine learning approach is robust, rapid, precise, and can potentially be used as a non-invasive clinical diagnostic tool for point-of-care applications.

This study is very interesting, timely and vel planned. It describes a proof-of-principle method for pathogen identification via VOC profiling using DART-SPME samples as a quick and accurate tool for detection of infection. PCA analysis and ROC and PRC results confirmed choosing ran appropriate research model. Authors proved that the ML algorithms are able to automatically identify peak locations in the MS data that have discriminatory importance. I agree with authors that POC tools employed in the proximity of patient care are essential to expedite patient care and decrease reliance on slower diagnostic techniques to identify pathogens and their associated antibiotic resistance. The research proposed in this study is identifiable a pathogen based on its VOC profile.
